# Caftaric Acid Isolation from Unripe Grape: A “Green” Alternative for Hydroxycinnamic Acids Recovery

**DOI:** 10.3390/molecules26041148

**Published:** 2021-02-21

**Authors:** Veronica Vendramin, Alessia Viel, Simone Vincenzi

**Affiliations:** 1Centre for Research in Viticulture and Enology (CIRVE), University of Padova, Viale XXVIII Aprile 14, 31015 Conegliano (TV), Italy; veronica.vendramin@unipd.it (V.V.); alessia.viel@unipd.it (A.V.); 2Department of Agronomy, Food, Natural Resources, Animals and Environment (DAFNAE), University of Padova, Viale dell’Università 16, 35020 Legnaro (PD), Italy

**Keywords:** hydroxycinnamic acids, caftaric acid, verjuice, FPLC, unripe grape juice

## Abstract

Phenolic acids represent about one-third of the dietary phenols and are widespread in vegetable and fruits. Several plants belonging to both vegetables and medical herbs have been studied for their hydroxycinnamic acid content. Among them, *Echinacea purpurea* is preferentially used for caffeic acid-derivatives extraction. The wine industry is a source of by-products that are rich in phenolic compounds. This work demonstrates that unripe grape juice (verjuice) presents a simple high-pressure liquid chromatography (HPLC) profile for hydroxycinnamic acids (HCAs), with a great separation of the caffeic-derived acids and a low content of other phenolic compounds when compared to *E. purpurea* and other grape by-products. Here it is shown how this allows the recovery of pure hydroxycinnamic acids by a simple and fast method, fast protein liquid chromatography (FPLC). In addition, verjuice can be easily obtained by pressing grape berries and filtering, thus avoiding any extraction step as required for other vegetable sources. Overall, the proposed protocol could strongly reduce the engagement of solvent in industrial phenolic extraction.

## 1. Introduction 

Phenolic acids constitute about one-third of the dietary phenols and are widespread in vegetable and fruits. Phenolic acids are divided into two subgroups, the hydroxybenzoic (HBAs) and hydroxycinnamic acids (HCAs). HBAs show a C6–C1 structure and include gallic, *p*-hydroxybenzoic, protocatechuic, vanillic, and syringic acids, while HCAs are characterized by an aromatic ring with three-carbon side chain (C6–C3) and are primarily represented by caffeic, ferulic, coumaric, and sinapic acids (Figure 1). 

In the past years, HCAs gained attention because of their cosmetic application as anti-tyrosinase, anti-collagenase, and anti-hyaluronidase activity, apart from an interesting photo-protection action [1], and for their possible application as a food additive to prevent oxidation [2]. HCAs are key precursors of several more complex polyphenols, are structural components of the cell wall, are involved in the plant defense system, and act as signaling molecules [3]. In plants, the first HCA produced is the *p*-coumaric acid, which is obtained from phenylalanine or tyrosine. This is then transformed into caffeic acid by hydroxylation. Ferulic and synaptic acid derive from caffeic acid by methoxyl and hydroxyl substitution and, in the case of synaptic acid, from an additional methylation [1]. HCAs are found in several conjugated forms, including amides (conjugated with mono- or polyamines, amino acids, or peptides), esters (mainly esters of hydroxy acids, such as tartaric acid and sugar derivatives), and sugars. Cinnamate esters occur widely in higher plants, while the amides seem to be less present [4]. Caffeic acid (CA)-derivatives is a group of compounds derived from modification of caffeic acid by esterification with organic acids, such as quinic acid (i.e., chlorogenic acid, neochlorogenic acid), glucaric acid (caffeoylglucaric acid) [5], and more frequently with tartaric acid (i.e., chicoric acid [6], caftaric acid, and coutaric acid) [7]. To date, several plants have been studied for their CA-derivatives content, plants belonging to both vegetables and medical herbs, e.g., *Echinacea purpurea* [8], *Cichorium intybus* [9] and *C. endivia* [10], *Lactuca virosa* [11], *Eupatorium perfoliatum* [5], and *Smallanthus sonchifolius* [12]. Nevertheless, other potential sources are known, for example, CA-derivatives were discovered to be quite abundant in fruits and particularly in grape berries [13]. The wine industry is a source of by-products that can be exploited for the recovery of high value compounds. Pomace, lees, and canes have been well explored in the past [14,15,16] as polyphenols sources, while less attention was dedicated to other products, such as unripe grapes. Cluster thinning is a common practice used to avoid over-cropping in compliance with the production regulations and is deemed a way to accelerate ripening and increase the grape quality, even if this function is still being debated [17]. Several parameters, among others, the grape variety and the weather conditions, which affect the plant production, define the degree of thinning, which could achieve up to 50% of cluster reduction [18]. Two main forms of berries reduction are spread, namely cluster thinning, consisting of the elimination of several complete clusters, and berry thinning, which involves the removal of the tips of the clusters [19]. Additionally, thinning is encouraged from institutional organs in particular cases (called “green harvest”), i.e., for a great imbalance between supply and demand of the international market [20], as in the case of the pandemic disease COVID-19. The clusters (or their parts) are usually left on the ground, and it makes it difficult to have a clear panorama of the effective waste mass. An alternative use of unripe grapes is the production of verjuice, an acidic juice traditionally produced in the Mediterranean area, which is extracted from the mechanical pressing of unripe green grapes. Verjuice has been mainly studied for its physicochemical and sensory properties [21,22], while its bioactive compounds were just recently isolated [23,24]. In grape berries, HCAs are constituted by caffeic, coumaric, and ferulic acids and are present mainly in their ester forms, associated with tartaric acid giving caftaric (CFA), coutaric (CUA), and fertaric (FEA) acids, respectively [25]. HCAs’ levels in the juice of different *Vitis vinifera* and *V. labrusca* varieties were recorded as very variable, namely 4339.9–1681.0 and 4154.5–786.7 μg/100 g in the former and in the latter, respectively [26]. The authors identified a strong difference in CFA content depending on *Vitis* species and varieties, while the other HCAs showed different patterns of accumulation amongst varieties, evidencing a general independency of hydroxycinnamic acids metabolisms. Total hydroxycinnamates concentrate mainly in grape berries pulp, and it has been recorded to peak prior to véraison [27]. The successive reduction of HCAs concentration depends on the grapes volume increase and on the engagement of key precursors into the biosynthesis pathways of other phenolic compounds [18,20]. Furthermore, HCAs reduce further during grape juice processing and winemaking, as these compounds are promptly oxidized by endogenous tyrosinase when the grape berries are crushed. Instead, the harvest of green berries and their processing through crushing and maceration could implement HCAs content [24]. Therefore, unripe grape juice (verjuice) represents a rich source of hydroxycinnamic acids and should be considered as raw material for the HCAs extraction. 

Several patents aiming to improve the recovery of CA-derivates from plants have been deposited in the past. Among them, the proposed raw material, *E. purpurea* adventitious roots, were recognized as the most suitable for the HCAs production at industrial scale because of their easy management and the high yield [28]. HCAs are extracted mainly in their ester form as chicoric (CCA), caftaric (CFA), and chlorogenic acids (CLA) [29]. Generally, these protocols involve the use of organic solvents, acidification, centrifugation, and the retrieval of final compounds by filtering or, more often, by separation with macroporous adsorption resin, which strongly improved the final HCAs purity, which could pass from 31% up to 72% *w/v* [28,30,31]. Finally, the compounds are further concentrated by crystallization upon acidification of the extract and its cooling or by evaporation at high temperatures (about 90 °C) that could, however, degrade HCAs. This last concentration step permits the achievement of purity values above 90% *w/w* [21,26]. 

To obtain pure compounds, due to the coexistence of different HCA in *E. purpurea*, an additional step of high-pressure liquid chromatography (HPLC) separation is necessary. Thus, those procedures led to the production of high volume of pollutants and the requirement of high-pressure liquid chromatography dramatically increase the process costs. 

Cluster thinning is commonly applied in different wine production regions, and the unripe berries are today underutilized, so this work proposes a method to valorize unripe grape by their juice as source of HCAs, with a particular attention paid to caftaric acid, which is supposed to have several healthy functions [32]. Verjuice obtained by grape berries manual pressing of five varieties, both international and local, have been compared during four successive weeks between bunch closure and early véraison, revealing that, overall, the highest amount of HCAs is recorded in the premature varieties and at the bunch closure. Additionally, a low-environmental impact chromatographic method that permits the reduction of chemical waste by eliminating the several purification steps has been tuned to separate and recover high purity caftaric acid from verjuice.

## 2. Results and Discussion 

### 2.1. Caftaric Acid Concentration over Green Grape Berries Maturation 

First, for a complete overview of the potentiality of the unripe grape as a caftaric acid source, it was considered important to determine which varieties and which moment of ripeness optimized CFA recovery. Therefore, five varieties, three international and two of the most important Italian ones, have been monitored on CFA production from bunch closure to the early véraison (Table 1).

Because caftaric acid concentration is affected by the berry volume increase during maturation, the concentration was adjusted by mass/juice yield and the CFA per kilo of fresh grapes weight (FW) was compared (Figure 2a). Analyses of variance performed on a linear model of the standardized CFA concentration data recognized significant effects of variety (F(_4,52_) = 23.2, *p* > 0.01) and date of sampling (F(_3,52_) = 17.8, *p* > 0.01), while color, as well as time request for maturation, did not statistically affect CFA content, differently from the data reported by Burin and colleagues at the grape technical maturation [26]. However, the interaction between the two main factors was statistically significant, and this suggests that different caftaric acid accumulation is observed depending on the grape variety. Indeed, Figure 2a highlights that not all the varieties were significantly affected by the week of sampling. Moreover, while in Pinot Noir (PN), CFA is strongly reduced between the first and the second week (85 mg/kg in one week), in Merlot (ME), the major reduction was recorded later, between the second and the last week of sampling 30.17 mg/kg (Figure 2a). PN and Glera (GL) revealed the greatest caftaric acid accumulation among red and white varieties, respectively (achieving 412.10 ± 12.28 and 298.86 ± 7.55 mg/L of CFA at the bunch closure). HCAs accumulation is influenced by grape light exposure [33], which is correlated with leaf surface, characters varying among varieties. Concerning CUA, the overall analyses of variance revealed a significant effect of the variety (F(_4,52_) = 15.745, *p* > 0.01) and the sampling date (F(_3,52_) = 3.007, *p* = 0.04) and again the interaction of the two factors was significant. Figure 2b shows that Sangiovese (SG) and Chardonnay (CH) were the major producers of coutaric acid, particularly at the bunch closure, followed by PN. If the total amount of HCAs is considered, it results that the highest amount is accumulated at (or before) the bunch closure, in decreasing order in PN (achieving 241.70 mg/kg), SG (182.58 mg/kg), CH (178.73 mg/kg), GL (161.96 mg/kg), and ME (102.67 mg/kg). Maturation variables, i.e., acidity, juice yield, and sugar content (SC), were related to HCAs concentrations (standardized per 1 kg FW) and revealed negative significant correlation of CFA with yield (r = −0.67, *p* > 0.01, *n* = 20) and SC (r =−0.48, *p* = 0.045, *n* = 20), while no significant correlation was found for CUA. Interestingly, the correlation between CFA and CUA results in a positive, not significant, correlation (r = 0.31, *p* = 0.17, *n* = 20).

### 2.2. Hydroxycinnamic Acid Esters in Verjuice 

The natural content in HCAs of unripe grape juice (verjuice) was analyzed using the HPLC method. All the analyzed verjuice revealed a simple peak profile for HCAs, with a great separation of the caffeic-derived acids and a general reduction of other phenolic compounds when compared with *E. purpurea* aerial part (Figure 3a,b). However, because of its availability, a verjuice obtained by pressing Riesling grapes collected in 2020 (BBCH stage 79) was used for purification. Two peaks were well distinguished. The first with a retention time (RT) of 8.54 min represented 74.1% of the total peak area, and a second peak at RT = 9.82 min corresponded to 9.6% of the total area (Figure 3a). 

Peaks identification was performed using commercial standards (CFA, RT = 8.54 min, caffeic acid, RT = 11.0 min, and coumaric acid, RT = 12.4 min), while coutaric acid (CUA), which is known to be the second hydroxycinnamic ester in grape for abundance [35], was identified by the comparison of HPLC profiles before and after an enzymatic degradation of ester bounds. The enzymatic reaction induced the appearance of two new peaks, corresponding to caffeic acid and coumaric acid. Additionally, the two original peaks were partially degraded, corresponding to the identified CFA peak and to the peak 2 (Figure 3a), which was consequently assigned to CUA. The analysis did not detect the fertaric acid that was probably present in too low concentration. The comparison between unripe grape juice and *E. purpurea* spectra made clear that while the latter is generally considered the best vegetable matrix for chicoric acid extraction, verjuice should be considered the best raw matrix for caftaric acid as well (Figure 3a,b). HCAs were quantified by the comparison of sample peaks area with a calibrating curve prepared using 25 to 200 mg/L of commercial CFA. Thus, CUA was expressed as CFA equivalents. Riesling grape juice contained 286 mg/L of caftaric acid and 38 mg/L of coutaric acid, namely about three times the maximum HCAs content detected in commercial verjuice [36]. This result could be explained in light of the strong effect of varieties and grape maturation point and because of the easy oxidation of HCAs during commercial verjuice preparation [37]. Considering the average yield of verjuice about 57% *v/m*, the data could be easily transformed into 163.02 and 21.66 mg/kg of fresh grapes, respectively, not dissimilar from the data reported for grape pomace by Kammerer et al. [38]. In the work of Wu and colleagues [29], several conditions were tested in order to evaluate which ones optimize the CA-derivates extraction from *E. purpurea* roots. Authors reported that, growing the adventitious root at 20 °C in an industrial system, 65 g fresh material could be obtained from 1 L of growth medium, corresponding to 10.4 g of dry material. The measured amount of caftaric acid was 4.9 mg/g of dry material corresponding to 784 mg per kilo of fresh roots. Therefore, verjuice could represent a promising source of caftaric acid for its easy preparation that avoids the additional costs of a specific industrial plant. 

### 2.3. Fast Protein Liquid Chromatography Applied to Hydroxycinnamic Acid Esters Separation

The most critical step in HCAs extraction from grapes raw material is represented by the isolation of the phenolic acids from other polyphenols. Chromatography was demonstrated to represent a handle tool for the selective isolation of HCAs ester in grape [39], and several methods have been tuned to obtain polyphenols high resolution peaks from fruits juice [40].

In addition to the traditional HPLC methods, Maier and colleagues [41] developed a method for CA-derivative esters recovery from ripe grape pomace based on the high-speed counter-current chromatography (HSCCC). This chromatography allowed the extraction and to successful separation of caffeic acid, coumaric acid, and ferulic acid esters by the head-to-tail elution mode, where the target compounds were separated from co-extracted polyphenolics and subsequently isolated in a second run. Liquid chromatography required a significantly longer time for separation; thus, CA-derivates separation required up to 390 min for the elution in the second HSCCC run. Additionally, this method involved the preliminary extraction with methanol and ethyl acetate and two mobile phases based on a mixture of hexane/ethyl acetate/methanol/water 3:7:3:7 (*v*/*v*/*v*/*v*) and tertbutyl-methyl ether/acetonitrile/*n*-butanol/water, 2:2:1:5 (*v*/*v*/*v*/*v*), both acidified by 0.5% of trifluoroacetic acid (TFA), which represent high pollutant waste. 

The simplicity of phenols profile of verjuice made possible a handy sample manipulation and the use of low-pressure chromatography as separation technology. Filtered juice of unripe berries has been processed without any sample preparation. After some preliminary tests, it has been determined that 50 mL of verjuice was the uploading limit for a column volume of 20 mL. Nevertheless, this limit could be easily overtaken by rearranging the column sizes.

Separation was monitored by means of the UV detector (at 280 nm). After sample loading, the column was washed with deionized water plus 0.5% trifluoroacetic acid (TFA) to remove the unanchored compounds, and then the target molecules were eluted by gradient of water: alcohol that achieves 30% *v/v* of alcohol in 100 min. 

Two commonly used solvent have been tested for the fast protein liquid chromatography (FPLC) separation, namely methanol, which is commonly used in HCAs chromatographic analyses [35], and ethanol, which was considered more suitable for further food application. The chromatographic profile revealed that well defined peaks could be obtained by methanol elution (Figure 4a), while ethanol elution evidenced less separation capability (Figure 4b). 

Then, the methanol protocol was used in ten successive sample loadings, which obtained a repetitive elution profile. The first peak was assigned to CFA by HPLC analyses of its fractions. All the fractions that contained CFA at the minimal purity of 98% have been collected and freeze-dried. The final amount of crystallized CFA was 82 mg, which means a potential of 93.48 mg of compound obtained from 1 kg of fresh grapes if a verjuice yield of 57% *v/m* is considered. As previously demonstrated [41], high-speed counter-current chromatography (HSCCC) leads to the recovery of high pure CA-derivates, i.e., 97.0% for CFA, 97.2% CUA, and 90.4% for fertaric acid. The method here proposed achieves similar results in terms of CFA purity with a strong reduction of solvents and time; indeed, the HSCCC method permitted the isolation and recovery of 6 mg of caftaric acid, from 10 g of freeze-dried pomace, after a preliminary extraction that required 800 mL of methanol/0.1% HCl *v/v* and 400 mL of ethyl acetate followed by the compounds separation in about 120 mL of hexane/ethyl acetate/methanol/water 3:7:3:7 *v/v/v/v*/ 0.5% TFA plus 40 mL of ether/acetonitrile/n-butanol/water, 2:2:1:5, *v/v/v/v*/ 0.5% TFA, while in this new method, 8.2 mg of caftaric acid is obtained by the direct separation of 50 mL of verjuice in 70 mL of methanol 1:6 *v/v*/ 0.1% TFA. 

## 3. Materials and Methods

### 3.1. Materials and Sample Preparation

Unripe grapes of five varieties, namely Pinot Noir (PN), Chardonnay (CH), Merlot (ME), Sangiovese (SG), and Glera (GL), were collected in the experimental vineyard of “Scuola Enologica di Conegliano G.B. Cerletti” (Treviso, Italy) in four successive weeks of 2019, between stage 73 and 83 of the BBCH scale. Samples were promptly added with 0.2 g/kg of potassium metabisulphite and processed in a basket press. The obtained juice was centrifuged at 2000× *g* for 5 min, then vacuum filtered through 1.6 µm glass fiber filters (VWR, Milan, Italy) and kept frozen until HPLC analyses. 

Additionally, unripe grape juice obtained from Rhine Riesling harvested at the véraison stage in 2020 was used for hydroxycinnamic acids recovery. Grape clusters were destemmed and washed before pressing with a small-scale stainless steel basket press. The basket press was loaded with berries in presence of 0.2 g/kg of potassium metabisulphite. The juice was centrifuged and filtered as described above and used for HPLC analyses and FPLC immediately. All reagents were analytical grade and were purchased from Sigma (Milan, Italy) unless otherwise stated. Chromatographic identification and quantification of caffeic acid, coumaric acid, and caftaric acid (CFA) were performed by the comparison of Riesling verjuice peaks with their commercial standard, while coutaric acid (CUA) was identified after juice enzymatic treatment. CFA standard curve was used for the quantification. The enzymatic treatment was performed using a commercial pectolytic enzyme with cinnamoyl esterase secondary activity. Verjuice (10 mL) was treated with 10 g/hL of enzyme and kept for 30 min at room temperature (25 °C) until the end of the reaction. Then, the sample was treated as described above before the injection. 

### 3.2. Grape Degree of Maturation Parameters 

Verjuice was immediately characterized by sugar content (SC) and total acidity (TA). Sugars were enzymatically determinaed by Hyperlab automatic multi-parametric analyzer (Steroglass, Perugia, Italy) by means of enzymatic kits, while titratable acidity was measured according to the official methods of wine analysis (Commission Regulation (EC) No1293/2005 of 5 August 2005 amending Regulation (EEC) No2676/90 determining Community methods for the analysis of wines).

### 3.3. HCAs Determination in High Performance Liquid Chromatography (HPLC)

Hydroxycinnamic acids (HCAs) separation was performed by C18 Lichrospher (4 × 250 mm, 5 µm, Agilent Technologies Italia, Milan, Italy) using a 1525 Binary Pump (Waters, Milan, Italy) equipped with 2487 Dual Band Absorbance Detector (Waters, Milan, Italy). Freshly prepared verjuice was centrifuged and filtered (0.2 μm), then it was injected (10 μL) and analyzed using the method proposed by Vanzo and colleagues [42] with modifications. Mobile phase was kept as proposed by the authors, while the flow rate was raised to 0.6 mL/min and the gradient was modified as follows: (A) Milli-Q water and 0.5% of formic acid *v*/*v* and (B) gradient-grade methanol and 2.0% of formic acid *v*/*v*. The gradient program was 0 min, 16% B; 7 min, 50% B; 8 min, 100% B; 8–12 min, 100% B; 13 min, 18% B; and 13–18 min, 18% B. The column temperature was kept at 40 °C. Hydroxycinnamic acids and esters were detected at the wavelength of 280 nm for purity determination and 330 nm for HCAs quantification; the peak areas were analyzed by software Breeze Version 3.3 (Waters, Milan, Italy). 

### 3.4. HCAs Retrieve by Fast Protein Liquid Chromatography (FPLC)

Filtered Riesling verjuice (50 mL) was loaded onto a Bio Scale Column MT20 (15 × 113 mm, internal volume 20 mL, Bio-Rad Laboratories, Milan, Italy) packed with LiChrosorb RP-18 (Sigma-Aldrich, Milan, Italy) and connected to an FPLC (AKTA purifier 10). The column was previously equilibrated with deionized water with 0.1% trifluoroacetic acid (TFA) and, after the sample loading, the column was washed with the same buffer to remove unbound sample components. The target compounds were eluted with a gradient of methanol 0.1% TFA, which linearly achieved 30% in 100 min with a flow rate of 2 mL/min. Fractions of 3.5 mL were collected by means of a fraction collector. The elution was monitored by recording the signal at 280 nm, and the purity was checked by HPLC analysis. Fractions containing at least 98% of CFA were pooled together and freeze-dried by Heto cooling trap (Analitica De Mori, Milan, Italy). 

### 3.5. Statistical Analyses

R software (R version 3.0.1) was used for statistical analysis. Differences were evaluated by one-way ANOVA and the Games–Howell post-hoc analyses. Variable relationships were tested using Pearson correlation. Statistical significance was attributed with *p*-value < 0.05. 

## 4. Conclusions

Hydroxycinnamic acids and their derived ester gained new attention recently in light of their potential application as antioxidants and as bioactive molecules in food and cosmetic formulations. 

Nowadays, hydroxycinnamic acids are mainly extracted form *Echinacea purpurea* roots, which are cultivated in an industrial plant set up with airlift bioreactors and require strictly controlled conditions of light, temperature, and nutrient availability, conditions that determine high cost of management. Nevertheless, other vegetables and herbs represent rich sources of HCAs and among them, grape berries. 

In general, the extraction of HCAs from grapes’ raw material, such as grape pomaces, faces the main problem of HCAs isolation from other phenolic compounds. On the other hand, verjuice polyphenols consist of a major part of hydroxycinnamic acids. This allows the reduction of costs and time for extraction and separation; the method here proposed demonstrates that a low-pressure separation procedure using fast protein liquid chromatography (FLPC) can be easily used to obtain high purity caftaric acid. 

This work proposes the unripe grape juice as a new source of hydroxycinnamic acids, mainly represented by caftaric acid. This new approach gives two important technological advantages: the valorization of vineyard by-product in place of the installation of industrial plant for specific raw material production and the possibility of a more handy isolation of the target molecules. It should be underlined that this solution meets the general requirements of a new low-environmental impact alternative toward the reduction of solvents and the simplification of pure molecule recovery. 

## Figures and Tables

**Figure 1 molecules-26-01148-f001:**
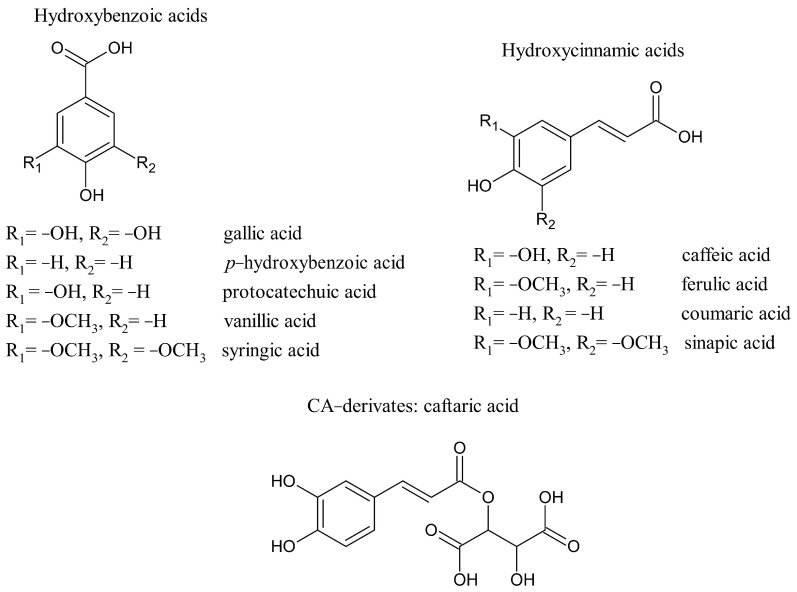
Chemical structures of hydroxybenzoic and hydroxycinnamic acids and of caftaric acid.

**Figure 2 molecules-26-01148-f002:**
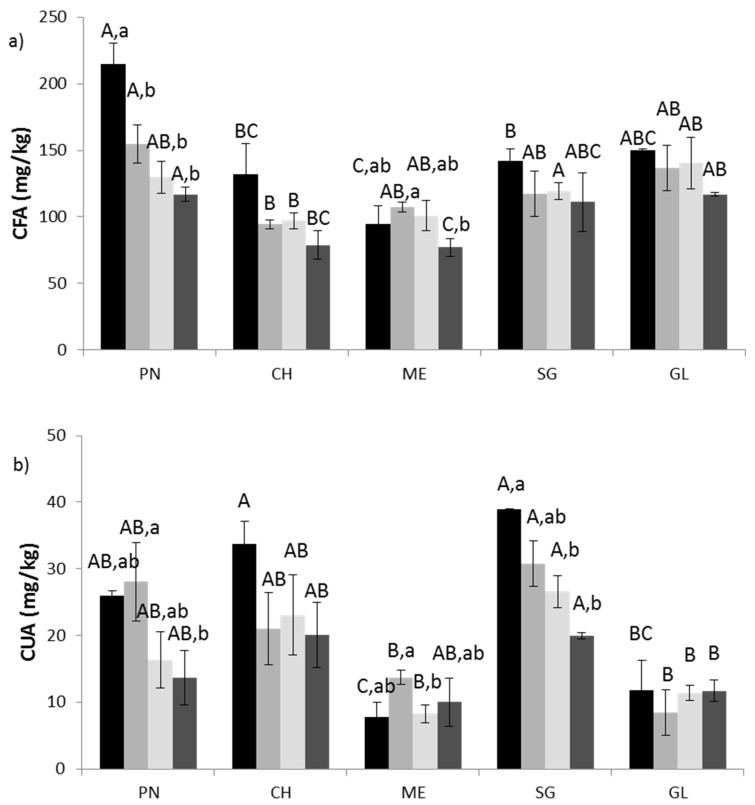
Standardized caftaric acid (**a**) and coutaric acid (**b**) content of verjuice. PN: Pinot noir, ME: Merlot, CH: Chardonnay, SG: Sangiovese, GL: Glera. Black bars: week one of sampling, grey bars: week two of sampling, light grey bars: week three of sampling, dark grey bars: week four of sampling. Different capital letter indicates significant differences among varieties at the same week of sampling; different lowercase letters indicates significant differences among weeks within the same variety.

**Figure 3 molecules-26-01148-f003:**
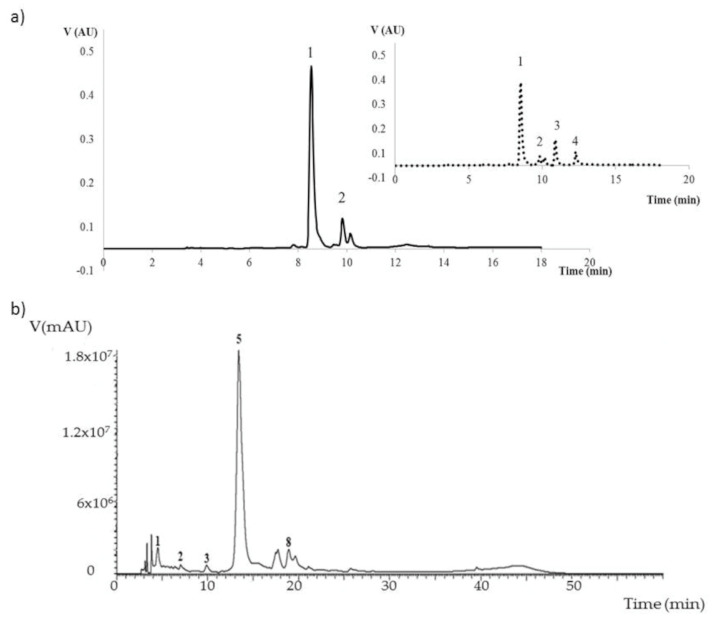
(**a**) HPLC profile of Riesling verjuice at 280 nm, before (continuous line) and after (dashed line) enzymatic treatment obtained in this work. Peak 1: caftaric acid, peak 2: coutaric acid, peak 3: caffeic acid, peak 4: coumaric acid. (**b**) HPLC profile (at 280 nm) of *E. purpurea* aerial part as reported by *Coelho* and colleagues [34]. Peak 1: caftaric acid, peak 2: 5-*O*-caffeoylquinic acid, peak 3: caffeic acid, peak 5: chicoric acid, peak 8: feruloylcaffeoyltartaric acid.

**Figure 4 molecules-26-01148-f004:**
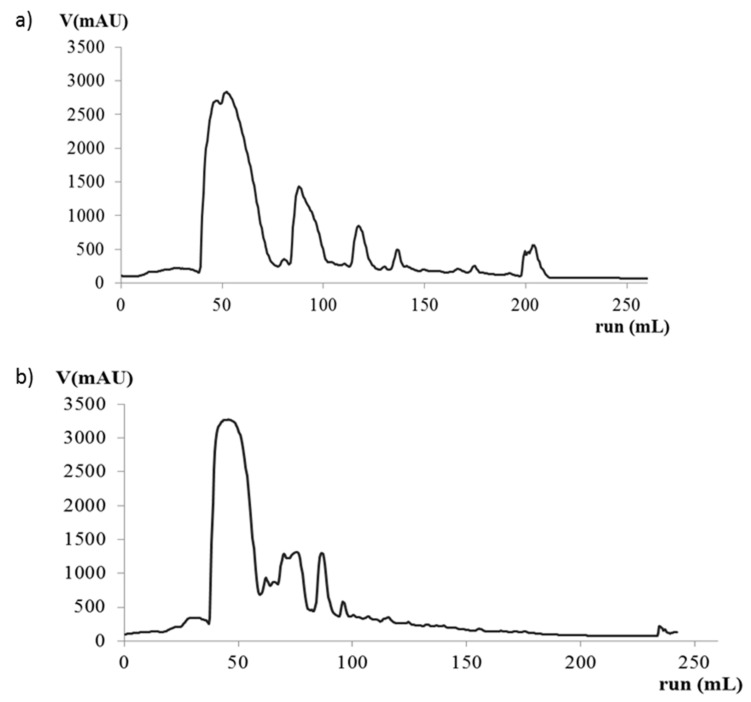
FPLCprofile of HCAs elution with (**a**) methanol and (**b**) ethanol as solvent.

**Table 1 molecules-26-01148-t001:** Yield, titratable acidity (TA), sugar content (SC), caftaric (CFA), and coutaric acid (CUA) concentration of verjuice obtained by different grape varieties during green berries maturation. Values represent the mean averages (n = 3) and standard deviation (in brackets).

Variety	BBCH Code	Yield (%)	TA (g/L) ^1^	SC (g/L)	CFA (mg/L)	CFA Purity (%) ^2^	CUA (mg/L)	CUA Purity (%) ^2^
CH	75	57.4 (0.3)	28.2 (0.4)	7.3 (0.6)	412.1 (12.2)	44.9 (3.1)	50.1 (2.8)	11.6 (1.3)
CH	77	57.3 (2.5)	35.1 (1.0)	8.0 (1.0)	230.4 (39.4)	46.8 (2.9)	58.8 (5.6)	11.5 (1.5)
CH	79	63.1 (1.6)	42.6 (3.6)	9. 7 (0.6)	173.1 (16.7)	48.7 (0.6)	14.3 (4.0)	11.1 (1.1)
CH	81	66.1 (1.0)	33.3 (3.6)	53. 7 (16.0)	257.5 (25.7)	47.3 (3.4)	70.5 (4.0)	11.42 (1.3)
GL	73	50.8 (1.1)	32.6 (1.3)	9.0 (0.1)	295.8 (7.6)	61.2 (3.0)	23.5 (9.3)	8.0 (0.2)
GL	75	46.0 (0.7)	41.8 (0.7)	11.7 (0.6)	272.5 (19.6)	67.6 (2.8)	50.2 (10.5)	6.36 (1.4)
GL	77	52.5 (0.9)	40.5 (0.7)	11.7 (0.6)	165.1 (1.4)	62.4 (7.0)	36.6 (7.7)	7.6 (0.5)
GL	79	55.1 (0.3)	39.5 (0.9)	27.0 (2.7)	179.8 (5.8)	63.3 (1.0)	23.0 (2.0)	7.3 (0.2)
ME	75	54.7 (2.8)	29.6(0.6)	8.3 (0.6)	207.0 (40.4)	52.4 (0.9)	54.2 (8.8)	7.2 (0.2)
ME	77	59.7 (0.9)	38.4 (0.1)	10.0 (0.1)	296.1 (34.1)	50.9 (4.1)	18.5 (7.6)	8.2 (0.2)
ME	79	62.3 (1.3)	42.5 (0.7)	11.0 (1.0)	222.8 (20.3)	58.3 (6.6)	28.1 (7.3)	6.2 (0. 8)
ME	81	64.9 (1.6)	34.8 (2.6)	36.3 (5.9)	154.0 (13.1)	48.7 (6.9)	36.5 (9.1)	7.5 (0.9)
PN	77	52.1 (2.3)	27.80 (0.6)	7.0 (0.1)	161.6 (15.9)	59.8 (3.4)	13.3 (2.0)	8.9 (0.6)
PN	79	56.7 (1.4)	38.9 (0.1)	7.7 (0.6)	211.1 (6.3)	58.4 (9.7)	47.2 (5.5)	11.7 (2.1)
PN	81	58.3(2.8)	45.7 (0.2)	21.0 (0.1)	267.4 (33.5)	63.0 (1.4)	21.9 (2.2)	8.2 (0.5)
PN	83	60.7 (1.6)	35.8(1.0)	43.7 (3.1)	192.9 (10.8)	57.8 (4.8)	22.6 (6.1)	8.1 (0.6)
SG	73	55.2 (2.2)	32.4 (1.1)	7.0 (0.1)	119.6 (17.6)	59.7 (4.9)	30.5 (7.4)	16.3 (1.5)
SG	75	57.1 (3.0)	36.5 (0.5)	8.0 (0.1)	118.9 (11.0)	56.3 (6.5)	15.5 (5.2)	13.6 (0.4)
SG	77	56.3 (2.4)	38.3 (0.1)	11.3 (0.6)	182.6 (30.8)	59.8 (1.7)	33.3 (3.1)	12.7 (0.1)
SG	79	60.7 (3.2)	33.7 (1.5)	36.3 (4.0)	212.4 (0.5)	50.8 (6.6)	21.3 (2.8)	12.6 (1.7)

^1^ Titratable acidity is expressed as Tartaric acid equivalents. ^2^ Hydroxycinnamic acid purity is calculated as the ratio between acid peak area and the total of peaks area at 280 nm.

## Data Availability

The data presented in this study are openly available in [repository name e.g., FigShare] at [doi], reference number [reference number].

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
