# Peer review of "Caftaric Acid Isolation from Unripe Grape: A “Green” Alternative for Hydroxycinnamic Acids Recovery"

_molecules, 2021, doi:10.3390/molecules26041148_

Round 1
Reviewer 1 Report
Many grammatical errors throughout, and a few spelling errors. A key spelling error is the use of "brunch" closure throughout, versus the correct "bunch" closure.
Figure 1 is very difficult to interpret and understand, especially the statistic analysis. A more detailed explanation of letters is needed in the text and/or figure legend, or the data should be reformatted in a clearer style.
The legend for figure 2 makes reference to part b), but no reference to part a) is provided.
The data presented in the last paragraph of the Results/Discussion section should also be presented as a table. It is not clear if this data was obtained by the authors as part of this study, or is references previously published data. If the former, proper citations need to be included. If the latter, the methods need to be described in the Materials and Methods section.
In the first paragraph of section 3.1, the authors state "Samples were promptly added with... ...and processes as described above." There is no "above" in the Materials and Methods, so the authors need to be more clear about what method is being referred to.
Additional clarification of the authors proposed practical use of their results needs to be included in the introduction and/or discussion. Is it a recommendation as a way to valorize otherwise useless waste (unripe grapes) from the wine-making industry? Or is this a recommendation that unripe grapes be harvested for the express purpose of recovering HCAs? Or are there other ideas?
Reviewer 2 Report
The justification for the research described in this manuscript is that there is a large "untapped" resource in vineyard waste that could be useful to the supplement industry. However, the authors don't provide any information regarding how much "waste" is generated in commercial vineyards during the period prior to verasion. While the authors show that the juice from unripe grapes is high in hydroxycinnamates no information is provided to support the claim of vineyard waste. Some number regarding fruit loss pre-verasion would help to support the purpose of this method and the use of verjuice. Please see attached for specific comments.

Round 2
Reviewer 1 Report
The revisions made significantly improve the manuscript.